# Understanding the Role of the Exodus in the Institutionalization and Dismantling of Apartheid: Considering the Paradox of Justice and Injustice in the Exodus

**Masiiwa Ragies Gunda**

New Testament Sciences, Institute of Catholic Theology, University of Bamberg, 96047 Bamberg, Germany; mrgunda75@gmail.com

**Abstract:** The Exodus played an explicit and implicit role in sustaining the policy and practice of apartheid in South Africa and in various other places that went through the pains of colonization. Interestingly, the same Exodus also played a central part in the resistance to and the subsequent dismantling of the apartheid policy and practice in South Africa. That readers on both sides of the divide found solace in the Exodus was put down to the common assumption that guided both parties. The assumption of historicity caused the Exodus to be read as if it were a photographic record of what happened and the experience of oppression and discrimination by the readers assigned the Exodus a historical status for speaking to a historical situation. The assumption of historicity was central in the destructive uses of the Exodus thereby creating a cycle of oppressed–oppressors across the African continent, as groups took turns to seek out their own advantage. An assumption of justice was proposed as an alternative guiding principle through which justice for all, in line with pivotal events of the Old Testament, can be realized in the world.

**Keywords:** Exodus; racism; justice; righteousness; assumption of historicity; assumption of justice





## 1. Introduction

In 1948, the National Party, which had won elections to govern South Africa, promulgated Apartheid as an official government policy. Apartheid is a compound word made up of "apart" and "heid", which is an Afrikaans suffix denoting "-ness" or "-hood". Apartheid, therefore, can be understood as "apartness" or "separateness". It was understood as the policy of "separate development" in which the government would enforce boundaries between the peoples of South Africa, especially based on race: White, black, colored, and Asian. This political policy remained in force in South Africa until 1994 (Manavhela 2009, p. 3). While apartheid was officially made a government policy in 1948, it has been observed by various scholars that it had started earlier within missionary churches in South Africa, especially among the Reformed Churches of South Africa, from which "theologians developed an impressive biblical case for apartheid-apartness. They were totally convinced that the Bible endorsed the separation of the races. They insisted that their policies were pleasing to God because they were grounded in Scripture" (Giles 2016). This, therefore, made apartheid not only a political ideology but also a religious ideology, maybe even a theology. It has been observed that "a biblical hermeneutical justification of apartheid was triggered by 'a sense of anxiety about the mixing of races, together with a concomitant uncovering of certain 'principles' on race in Scripture'" (Farisani 2014, p. 210). It is of particular interest for this study that the Bible in general was implicated in the rise and institutionalization of apartheid in South Africa. Through the use of the Bible, a systematic policy of racial superiority and physical separation of races emerged (Torevell 2019). This policy, while it did not become "official government policy" in the neighboring countries, was, however, silently put into practice by various churches and colonial governments in the region.

Gerald West (2008, pp. 101, 108) is right that "since its arrival in Southern Africa, the Bible has been a site of struggle . . . apartheid was built on the Bible, and so was the liberation struggle." In light of this observation by West, it is important to note that the dismantling of apartheid, like its rise, benefited greatly from the deployment of the Bible. Iconic figures, like Nelson Mandela, were revered as a type of biblical Moses, as they led the fight for liberation.

> "In 1960, ten leading Reformed Afrikaner theologians published a series of essays condemning Apartheid and the claim that the Bible endorsed racial separation. They were put on trial for heresy, found guilty, and denounced by the prime minister, Dr. H. Verwoerd, himself a theologian. In 1963, Beyers Naudé, another Afrikaner theologian, spoke out and wrote in opposition to the claim that the Bible supported Apartheid. Naudé and his family were completely ostracized by their fellow Afrikaners. He was forced to resign as minister and put out of his home without a salary". (Giles 2016)

The resistance to apartheid was constructed from several texts from the Bible, just as several other texts had been used to propagate and support apartheid both in the Church as well as in the country in general, clearly sustaining West's observation that the Bible has been a site of struggle.

This paper seeks to understand the role of the Exodus in the construction and justification of apartheid as well as in attempts to dismantle it. It is interesting to note that most of the theologians who supported apartheid rarely made direct quotations from the book of the Exodus, yet a closer reading of the arguments will establish an unmistakable echo of the Exodus narrative. However, there have been explicit references to the Exodus by those who feel and are oppressed (Pacheco 2017), showing how biblical texts have been appropriated by succeeding generations (Pederson 2012).

## 2. The Rise of Official Apartheid

It was noted above that apartheid existed in South Africa as both unofficial and official politico-religious ideology whose intention was to enforce the separation of the races. According to Richardson (1986, p. 4), "the term apartheid refers primarily to the policy of the governing National Party which has been in power in South Africa since 1948" and remained in power until 1994, when Nelson Mandela won the general elections. This section traces the rise of apartheid to become "official government policy" in South Africa. In essence, the 1948 political move to make apartheid government policy was not, in effect, taking the country in a new direction but can be understood, rather, as a culmination of almost a century of the development of apartheid as a central component of life in South Africa. During the subsistence of apartheid, "whites and blacks had to be separated by any means and under all circumstances" (Manavhela 2009, p. 3).

The earliest moves towards the adoption of apartheid in South Africa can be traced back as early as 1829, when some members in the Dutch Reformed Church requested for "racially separate communion services", a request that was rejected "according to the teaching of Scripture and the spirit of Christianity." This position was further reiterated in 1834. However, in 1857, the Dutch Reformed Church in South Africa (NGK)'s synod "recognized the Scriptural desirability of racially integrated worship, but reluctantly allowed racially separated places of worship because of the weakness of some" (Richardson 1986, p. 5). It is factual, then, to recognize 1857 as marking the beginnings of apartheid in South African Christianity. To actualize the 1857 resolution, the Dutch Reformed Church in Africa, also known as the Bantu Church, was established in 1859, for black converts, and in 1881 the Mission Church for "Coloured" (mixed-race) members of the NGK was established. Effectively, by the time the National Party government with its apartheid policy came into power in 1948, racial separateness within the Dutch Reformed Church had become the axiom. The last of the three "daughter churches," for Asian members, was established in 1951 (Richardson 1986, p. 5). In short, the policy of apartheid in South Africa, which was propagated by the National Party government, was a policy that had already been widely

accepted by most white Christians in the Dutch Reformed Church (and arguably in other churches, as well) (Joebgen 2016).

While it gained notoriety as a political ideology, there is no mistaking the theological and religious foundations on which this policy was founded. The central argument for apartheid was an "emphasis on the idea that what God has separated we should not put together" (Vosloo 2015, p. 197). In an environment where the desire was to keep the people separated by race (ethnicity, class etc.), biblical concepts such as election and covenant were appropriated to give a divine aura to the policy and practice of separating people on racial grounds. Apartheid and racism were provided with a "sacred canopy" (Richardson 1986, p. 4). Racial differences (and this could be applied to ethnic differences as well) were understood as divinely ordained hence not to be transgressed in line with several purity instructions scattered throughout the Bible. Promoting apartheid was, therefore, understood as defending the sanctity of the "gospel". Some Reformed theologians "argued that the Bible taught that humankind, by the will of God, was separated into different races that should each have their own lands. They insisted that Apartheid was pleasing to God because it was endorsed by Scripture" (Giles 2016).

Several biblical texts were cited in support of apartheid by theologians, among them Genesis 9 (the curse of Ham), Genesis 11 (the Tower of Babel), Deuteronomy 32:8 (the division of the peoples is by God), 1 Corinthians 7:17–24 (remain in the condition in which you were called), and Revelation 5:9 and 7:9 (which speak of people according to their nations or tribes). From these and other texts "conclusions were summarily drawn about racial segregation within society and in the church" (Farisani 2014, p. 211; Williams 2004, p. 155; Pederson 2012, p. 441). While biblical texts had been used within the confines of the Church as "separate worship and communion" was being debated, biblical texts became even more prominent in the public support of the official government policy from 1948, giving the government not only political authority based on an election outcome but also theological justification, making this government appear "divinely ordained", since its policy was based on biblical injunctions. Lubbe (2001, p. 98) indicates five strong points of the base of principles of Apartheid:

> That Scripture places the oneness of the human race (Genesis 1:26–29); that Scripture relates the division of humankind into races and nations to "an act of God" (Genesis 10–11; Deuteronomy 1:26–29; Acts 17:26); that Apartheid reaches over the whole sphere of life of the people—national, social and religious; that God's blessings rests on upholding Apartheid (Deuteronomy 7:11) and that a higher spiritual unity is established in Christ (Ephesians 4:4–6; Galatians 3:28 and Revelation 5:9).

While the Reformed Churches have been the most prominent churches behind the rise of not only an "apartheid theology" but also the political "apartheid ideology and policy", this by no means must be taken to mean all other missionary churches did not practice some form of apartheid because they did. In fact, apartheid in churches, not only in South Africa but across the African continent, had become an all-pervading heresy. Several efforts by these churches in putting out statements against apartheid were driven by a realization that they were also implicated. Obstacles were put in the way of non-whites making certain positions and some roles in these churches were put out of reach for non-white converts (Richardson 1986).

## 3. The Role of the Exodus in the Rise of Apartheid

What is observable from the preceding section is that the book of Exodus appears not to have been widely cited in the development of apartheid theology, ideology, and policy. Does this, therefore, mean that the Exodus played no part in the rise of Apartheid as a theological-ideology, which was turned into a government policy in South Africa in 1948? This section seeks to trace the manner in which the Exodus was used during this period and how it became a foundational narrative of what became a concerted effort to promote the otherness of non-whites during the apartheid era. While we noted the absence of texts from the book of Exodus in the textual references in defense of apartheid, the Exodus from

Egypt to Canaan, the Promised Land, was actually used in a significant moment in the rise of apartheid in South Africa. It is important to observe that Dutch sailors were responsible for setting up the Cape of Good Hope (Cape Colony), as a stop-over post for their journeys to the east. The Cape colony was ruled by the Dutch East India Company from its setting up in 1652 until 1795, at which point it came under British rule (1795–1803), was then given back to the Dutch (Batavia Republic controlled by France) in 1803, but was re-taken by the British in 1806 (Manavhela 2009). Under British rule, the Afrikaners or Dutch people that had settled in the Cape colony felt the pressure of the changes that were being introduced by the British. The English language became the official language of the colony, and some British laws were applied in the colony as well. Of special importance for this paper was the abolition of slavery, which had been promulgated by Britain in 1807 and was applied to the Cape colony around 1833. According to Ryrie (2017);

> *The last straw was the abolition of slavery throughout the British Empire in 1833. One Afrikaner later recalled her outrage at "the shameful and unjust proceedings with reference to the freedom of the slaves . . . their being placed on an equal footing with Christians, contrary to the laws of God and the natural distinction of race and religion." "Christian" was a tribal identity, "race-and-religion" a single word. For this Afrikaner, there was only one solution: "We withdrew in order to preserve our doctrines in purity." A pilgrimage: the so-called Great Trek of the late 1830s, which saw around twelve thousand Afrikaners leave the Cape Colony and move north and east. The DRC's synod denounced the trek and banned its ministers from joining, but the "Voortrekkers," undaunted, knew that they were leading their covenanted people from British captivity into the Promised Land.*

In the Great Trek, the Exodus came alive! "This trek was somehow identified with the exodus of Israel in the Old Testament since they claimed to liberate themselves from British occupation" (Strauss 1994, p. 95). The imagery of an oppressive Pharaoh, in this case the British, and an oppressed people, in this case, the Afrikaners, was invoked. "The British were the main source of threats for the Boers [Afrikaners]. The English were more powerful than them. The Boers were repressed; they felt their privileged position, of being white, was at stake. They wanted to protect their endangered position from the British by shifting the repression and vent their anger to the black people. The threats were shifted to the black people whom they thought could chase them out of their country" (Manavhela 2009, p. 23). The escape from the British was a divinely inspired exodus from tyranny to liberation. What was at stake was the desire to "preserve pure doctrine", that is apartheid. In invoking the Israelite exodus, the Afrikaners extrapolated "Israel's insight into the redemptive character of Israel's God expressed in Israel's constitutive exodus experience. Israel's God liberated the exploited, demonstrating favor toward the downtrodden. To exploit, then, is to fail to reflect God's character as God's people; it is to align with those who oppose God's purposes; it is to become Pharaoh" (Biddle 2011, p. 126). This understanding had no contradictions with the ongoing enslavement and racial discrimination of blacks because apartheid itself was God's design. Oppression, that called for God's intervention and that necessitated the Great Trek exodus, mattered because it was being perpetrated against the white race otherwise inter-racial oppression was considered part of God's plan in the light of the "curse of Ham" (Gen. 9).

The power of the exodus was in its historicity! A history that was now being repeated. The exodus was physical in the sense of being a movement from one physical space to another, it involved the displacement of the indigenous occupiers of the "Promised Land" because they were occupying land that had been promised to the invaders. There was, therefore, no oppression or injustice committed against the indigenous people, their fate had been sealed by God. Oppression was that which the invaders were escaping from.

What is particularly instructive is the manner in which the exodus played a foundational role in instilling or justifying racist tendencies among some white people as they escaped or left their homelands for new lands. Yet the same exodus did not provide a foundation for the realization of the faults in apartheid theology and ideologies—the central exodus motifs of liberation, justice, and freedom were relativized to apply only to

whites and not to blacks or other indigenous peoples. This was true of South Africa and all other colonies that went through one form or another of apartheid and its racial tendencies.

Because people are usually more keenly aware of their own victimization than of the privileges they accrue by exploiting others, most users of the Exodus story identify with the slaves, not the pharaoh. The Exodus story's master–slave dichotomy does not foster an awareness of being mistreated based on one identity while being in an oppressive role based on another. The typologies imposed on the story encourage groups to regard themselves or others as occupying only one position—either oppressed or oppressor.

This exact observation can be applied to the Afrikaners who did not see themselves as victims and perpetrators of oppression, injustice, and racial discrimination. They were "permanently the fleeing Hebrews" never oscillating between "Hebrew slave and Egyptian Pharaoh." "The single most persistent image . . . is that of the chosen people, in reworking the trope of chosenness" (Pederson 2012, p. 441). In this appropriation of the Exodus, it is highly probable that the story of Moses and the Israelites was more meaningful to whites assured of their own privilege than to blacks struggling against institutionalized racism. The laws instituted and promoted by the white-Hebrews promoted

> *in the mind of the smallest white child the conviction of First by Birth, eternal and irrevocable, like the place assigned to the Levites by Moses over the other tribes of the Hebrews. Talent, capabilities, nothing has anything to do with the case. Just FIRST BY BIRTH. No one of darker skin can ever be considered an equal. Seeing the daily humiliations of the darker people confirms the child in its superiority, so that it comes to feel it is the arrangement of God. By the same means, the smallest dark child is to be convinced of its inferiority, so that it is to be convinced that competition is out of the question, and against all nature and God.* (Pederson 2012, p. 441)

What this section has done is highlight how, even when it was not explicitly invoked, the Exodus was a foundational experience and narrative that sustained apartheid theology and ideology with its emphasis on the occupation of the Promised Land, its identification of peoples in the categories of "chosen" and "not chosen". The concepts of liberation, justice, and freedom were leveraged against so-called "divinely ordained boundaries of race" implying that biblical liberation, justice, and freedom were for the "chosen" or "elect", the Israelites! In the context of South Africa, and Africa in general, this meant the "imperial and colonial settlers" who considered themselves the "chosen" as opposed to the heathen African. The African's position was that of servitude not liberation!

## 4. The Role of the Exodus in Dismantling Apartheid

In the previous section, we have highlighted that even without being explicitly invoked in the construction and justification of apartheid both in the Church and State, the exodus did play a foundational role. This section seeks to investigate to what extent the exodus contributed, if it did, to the dismantling of apartheid policy in South Africa and generally in Africa and in other places where exodus-inspired racial discrimination had reared its ugly head. In many instances of oppression and exploitation, the Bible has inspired voices of resistance, justice, and revolution that "speaking against the structures of sin that have been responsible for the mass production of poverty and hunger in the world is not being this-worldly, it is being Christian" (Gunda 2015, p. 170). The same could be said about the exodus narrative and experience, as it has been widely adopted and adapted by people suffering marginalization, exploitation, and racial discrimination, as in the case of apartheid. According to Schneiders (2006, pp. 100–1);

> *"The oppressed find in the biblical text resources for their struggle. Liberation interpreters, both the lay people participating in the reading and the scholars who have the liberation of the oppressed their primary academic agenda, read the biblical text through the lens of grinding poverty, rampant disease, premature death and socio-political powerlessness. In the text they find the assurance that their suffering is not willed by God but unjustly imposed by those in power and that God is on the side of the oppressed."*

Nelson Mandela, the iconic leader of the fight against apartheid, who was imprisoned by the apartheid government, insisted that fighting for liberation was his calling. That he was willing to die for this cause. "That personal creed has echoed through the intervening years, becoming part of the political manifesto of South Africa's blacks, and transforming Mandela into a larger-than-life figure, a Moses who blacks think will lead them to freedom" (Kraft 1990). That Mandela was seen as "a Moses" demonstrates how the exodus was being understood by oppressed black people of South Africa. The actions of the whites had put black South Africans in Egypt, a land of oppression and exploitation. However, the blacks identified Mandela as their own Moses who would lead them to liberation and freedom. What had happened in Egypt back then was going to happen again in South Africa because history has a way of repeating itself. Because it once happened, it means it will happen again.

Desmond Tutu was also prolific in looking at the quest for liberation through the lenses of the exodus. "From his own context as a Black church leader in present-day South Africa he knows that God sides with the oppressed in the liberation struggle. But nevertheless he finds it necessary to state that this is true because the Bible says so" (Loader 1986, p. 156). For Tutu, Exodus-based sermons had a dual function, where it was meant to "prove" that something similar to what black South Africans were going through had happened in the past and God had decisively taken a position to support the oppressed. Further, Tutu implied that because black South Africans were suffering and oppressed, their suffering was bringing the Exodus or the Bible back to life and their suffering would instigate God to take action on their behalf, as God had done for the Israelites. In an address to white students in 1981, Tutu presents God as "the Great Exodus God, who took the side of the oppressed" (Loader 1986, p. 157). Seeing the existential suffering of the oppressed around him, Tutu approaches the Exodus from this historical experience and equates that historical experience with what the Exodus historically narrated.

Allan Boesak's sermons during the years of apartheid were full of references, explicit and implicit, to the exodus and understood the liberation of black South Africans in terms influenced by the exodus (Boesak 1979, pp. 78–83; 1984, pp. 19–25). Unlike Tutu who was explicit on God being on the side of blacks, Boesak placed God on the side of the righteous people, irrespective of race. The God of the Exodus took the side of the righteous against the wicked. "The Exodus in South African black theology is not appropriated by the blacks against the whites as though God was on the side of the blacks. In the South African context God is on the side of the righteous, that is those who are motivated by the love of God and neighbor" (van Aarde 2016, p. 7).

Persons converted to Christianity and who continue to experience oppression, injustice, and discrimination from their societies, especially from other Christians, tend to initially disown the faith and its foundational resources, such as sacred texts, yet the resolution normally emerges through some form of reorganization and reinterpretation of their experiences and reality and aligning that to the experiences and realities narrated in the Bible. According to Gunda (2009, p. 80);

> *"When one nationalist Obed Mutezo was asked about the practice of segregation in many churches, when he held the view that the Church was good, his response was that 'indeed churches were practicing segregation but for him missionaries and not the Bible were the problem.'"*

Whereas initially the Bible would have been seen as a colonizing, oppressive text, once the oppressed started reading the Bible without the agency of missionaries, the Bible spoke clearly about the wickedness of injustice, oppression, and discrimination that the oppressed found fault in the missionary readings and not in the Bible itself. This became very true of the exodus experience as well.

According to Ela (1988, p. 103); "Throughout the whole of scripture, which can be seen as a re-reading of the exodus, God brings forth words and deeds, revealing a God who is the last refuge of his beloved people subjected to exploitation, violence and misery. 'To oppress the poor is to insult their creator' (Prov. 14:31)." While the oppressors had previously

identified themselves with the chosen, it was clear to the oppressed that the oppressors were the pharaohs while the oppressed were the "children of Israel", the "chosen ones". They were the ones who found legitimation and comfort from knowing that their own exodus from oppression and injustice was possible because God would intervene the same way God intervened for the Israelites.

When looking at the apartheid system, which actually happened in all colonized countries under different names and sometimes not as established policy, the status of the Bible, which had initially been presented as unequivocally in support of apartheid, was suddenly transformed from a book that sustained oppressive structures to a book that challenged oppressive structures. The Zimbabwean readers of the Bible, who had at some point believed the authority of missionaries as readers and to a certain extent "owners" of the Bible, woke up to find themselves not as mere objects but as the major subjects of the biblical story. In doing this, Zimbabwean readers practically engaged in the drawing of lines of connection between the biblical texts such as the exodus, prophetic literature as well as the ministry of Jesus and their context as a community under the rule of an oppressive system (Gunda 2010, p. 104; see also West 1999).

The Bible, in general, was understood as a liberating document and the exodus became a foundational drama that foretold the experiences of all oppressed peoples and how God was and would consistently fight in the corner of the oppressed. This is why the exodus has always been an important reference for liberation theology (Pacheco 2017, p. 55), as well as in the responses to apartheid.

In all this, the first step in undermining the legitimacy of apartheid, racial discrimination, injustice, and oppression was the identification with the Hebrew slaves with Egypt becoming the symbol of all forms of oppression and discrimination. Emboldened by their identification with the Hebrew slaves, the response moved into its second step in which an alternative gospel was developed, which explicitly denied the supposed divine ordination of apartheid hence The General Council of the World Alliance of Reformed Churches in 1982, (where racially discriminated delegates from the daughter churches of the Dutch Reformed Church refused to share Holy Communion with delegates of the mother church (Richardson 1986, p. 1)) promulgated;

> "We declare . . . that apartheid (separate development) is a sin, and that the moral and theological justification of it is a travesty of the Gospel and, in its persistent disobedience to the Word of God, a theological heresy". (United Nations Centre against Apartheid 1983, p. 3)

It is agreed by scholars that "The Kairos Document of South Africa hastened the end of apartheid by inspiring a new generation of conscientized and radicalized Christians to participate alongside secular liberation movements in the struggle to remove apartheid" (Draper 2008, pp. 39–40). Christian resources that had been used to justify apartheid were now fully invoked to undermine and challenge apartheid. Texts that had been central in the "gospel of apartheid" became central in proving the fault in the practice of apartheid. According to Gunda (2018, p. 51);

> "To begin with, from Genesis 1 to 18, there are several cases of human wickedness that anger God, suggesting that God had intended for peace, harmony, righteousness and justice among human beings created by God, seen especially, in the disobedience in the garden of Eden (Gen. 3), the murder of Abel (Gen. 4), the fornication between the sons of God and the daughters of men (Gen. 6) and the Tower of Babel (Gen. 11). However, these chapters suggest that left to their own devices, human beings were likely to forego justice and righteousness as the strong overpowered the weak. This made God, to set apart, an individual and line of persons through whom God's project of establishing a society based on righteousness and justice would continue. This is the background to the choosing of Abraham and in Genesis 18:19, God explicitly states the charge that is laid on Abraham."

Apartheid did not stand for justice, righteousness, or harmony among the different peoples, it legitimated injustice, exploitation, and racial discrimination of some people

based on the color of their skin. To that end, apartheid was to be understood as belonging to those forms of wickedness that anger God. The exodus was for exploited, oppressed, and discriminated people, not for the perpetrators of exploitation, injustice, and racial discrimination.

## 5. The Problem with the Methodological Assumption behind the Reading of the Bible in Sustaining or Dismantling Apartheid

In the world influenced by Christianity, the Bible has been given the status of an unrivalled historical source regarding the beginnings of the world. The 'Mosaic accounts' in the Pentateuch were considered historical and true and any other explanations were considered absurd and ridiculous (*Encyclopcedia Britannica* 1780, p. 3650). However, since the rise of historical critical approaches to the study of the Bible, "not only have historians stopped using the Bible as a major historical source, but we have also been witnessing a development where so to speak every new generation of authors of that particular scholarly genre most commonly referred to as 'The History of ancient Israel' have had their sources drastically curtailed" (Barstad 2008, p. 1). While the Bible might have lost its position as a historical source among critical biblical scholars, a historical assumption in the different approaches to the Bible has persisted and remains firmly in place. Whether the Bible is read by the ordinary believer in search of edification or a trained reader in search of understanding, there is a historical assumption that operates consciously or subconsciously, and this is apparent in the readings of the Bible in support of or in opposition to the practice and policy of apartheid.

The contestation surrounding what the Bible says about 'separate development' or apartheid betrays a commitment to a historicism that has now been deeply entrenched in our ways of thinking. The power of the Bible is seemingly dependent on the historicity of its claims in which case that which we consider historical is true and powerful while that which we consider not historical is fictional and powerless. If it is to be considered powerful then it must be believed that it happened otherwise it cannot be powerful. Whether trained or untrained, readers have this commitment overtly or covertly. According to Barstad (2008, pp. 14–15),

> "We are all 'brainwashed' by German historicism, and we have, according to this mental upbringing, a tendency to classify all written material in categories of true = historical, and not true = fictional. Did this thing happen, or did it not? Did Moses live, or did he not?"

The greatest challenge, therefore, has been the fear of being accused of denying the sacrality and power of the Bible by suggesting that there are elements in the Bible that are not history and cannot be read as if they were history. When it comes to the Bible, it is important to acknowledge that "the contents of the Bible are in large measure not a true historical source but are a religious ideology expressing itself in a form purporting to be historical narrative" (Barr 2000, p. 18). Whether it is the Primeval History, the Patriarchs, the Exodus, or even the Deuteronomistic History, what we have in the Old Testament are religious ideologies not merely histories. This must not impact the power of these narratives and poems negatively because the power of religious ideologies is not dependent on their historicity but on their capacity to shape the lives of believers.

This historical assumption is applied to the biblical texts, and it is used to accept the historicity of biblical claims at the level and magnitude asserted by the biblical texts or at a level considered reasonable by modern standards of historicity based on the contemporary remodeling of ancient history. When it comes to the Exodus, most pre-critical approaches to the study of the Bible accepted the magnitude of the Exodus of millions of people departing Egypt, while in historical critical approaches, the magnitude is revised to a few thousand people departing Egypt. Yet, most importantly, either approach assumes the historicity of the Exodus and places a premium on this historicity. The second dimension of this historical assumption is that the biblical text is considered historical if it appears to affirm the historical experiences of the reader. That those in support of or against

apartheid found the Exodus to be affirming their lived reality made the Exodus a historical narrative (see Hauser and Watson 2003). What we observe in this understanding of the historical assumption is that readers of the Bible, both pre-critical and critical, have resorted a hermeneutic of continuity that seeks to reconcile their current reality with the supposed historical reality of biblical times, hence helping the readers to "understand, apply, and respond to biblical texts" (Thiselton 2009, p. 1). "In the frame of an 'historical' retrospective, biblical passages are interpreted in a typological way . . . This typological exegesis refers events of the past to current events" (Maier 1996, pp. 125, 127). When it comes to the Exodus;

> *"The period of servitude of the Hebrew people in the land of Egypt narrated in the Exodus should not be thought of as a romantic story of heroism where God is the protagonist operating miracles, signs, and wonders. Rather, it must be thought of in the context of the pains and violations that are present in the condition of a slave, or in the context of the lives of those who are made slaves in colonized territory. This means that the historical context of the Hebrew people in Egypt bears similarities to the process of colonization undertaken by Europe that also reaches Brazil".* (Pacheco 2017, p. 56)

A direct appropriation was considered appropriate and justified because of the similar "historical context" between that of the Hebrew slaves and contemporary exploited peoples. "For captured Africans, America was no promised land . . . America is Egypt, its rulers terrible pharaohs, and before the Civil War, "Canaan referred not only to the condition of freedom, but also to the territory of freedom" (Kay 2008, p. 27). This historical assumption also influenced scholars of the Bible, who through the widespread adoption of the historical-critical approach in seminaries and universities, saw most scholars being pre-occupied with establishing the "historicity of the Bible". In the case of apartheid, both sides of the divide saw themselves as in continuity with the experiences of the biblical Hebrew slaves! This paradox was made possible by the historicization of truth by readers of the Exodus on both sides of the divide both sides found it difficult, if not impossible, "to respect and to understand that narrative truth is a different truth from that of conventional history, but that it is not a lesser truth" (Barstad 2008, p. 15). The power of the Exodus narrative is not in its historicity, but it is possibly tied to its ideological proclamations, that is, God detests exploitation, oppression, and injustice and will intervene on behalf of the oppressed.

### 6. An Alternative Methodological Assumption to the Exodus and How It Would Have Impacted the Development of Apartheid

In the preceding section, we have observed how the assumption of historicity played a critical role in determining how the Exodus was appropriated by readers of the Bible in support of or in opposition to apartheid. The full impact of this historical assumption is seen in the way in which one group could be liberative and exploitative simultaneously because the historical assumption was limited to the welfare of the reading group alone. The Afrikaners felt oppressed by the British yet they did not see the irony in their oppression of the indigenous peoples of South Africa. Similarly, the elites among the indigenous peoples invoked the Exodus against the colonizers yet ended up being oppressors of fellow indigenous people. "As far as the Biblical documents are concerned, it would be fallacious to interpret them only or primarily as historical documents. Many of the Biblical traditions do not even pretend to be historical documents in the usual sense of the word, that is, photographic records of what happened" (Roth 1963, p. 55). In this section, we propose an alternative assumption through which we could approach and appropriate the Exodus for the wellbeing of all.

> *"Rabbi Simeon bar Yohai taught: There is a story about men who were sitting on a ship, one of them lifted up a borer and began boring a hole beneath his seat. His companions said to him: 'What are you sitting and doing?' He replied to them: 'What concern is it of yours, am I not drilling under my seat?' They said to him: 'But the water will come up and flood the ship for all of us'. This story reflects upon an important and often misunderstood dimension of Israelite religion. I call this dimension corporate*

*responsibility, by which I mean the way in which the community as a whole is liable for the actions committed by its individual members. Corporate responsibility is an important concept because it is a fundamental theological principal in ancient Israel that God relates not just to autonomous Israelites, but to the nation as a whole. Inasmuch as God relates to the community as a whole, he holds each member of the nation to some level of responsibility for the errors of any other member of that community. Not only is one responsible for one's own proper behavior, but one must also actively prevent others from sinning".* (Kaminsky 1995, p. 11)

In line with this understanding of corporate responsibility, this is a principle that must be applied to humanity, as a whole. The Exodus, as a religious-theological ideology, has to speak to the reality of human beings and not to the reality of a select few. The assumption here is that of justice as opposed to that of historicity. The assumption of justice seeks to establish the role of religious-theological ideologies in the establishment of justice among all peoples, which means readings of the Bible, in this case, the Exodus, that privileges one group materially over others cannot be considered to have done justice to the religious-theological ideology nor to the human race. The fundamental problem with the readings based on the assumption of historicity was that biblical texts were taken as meaningless until they were made meaningful by the readers (Thiselton 2009, pp. 1–2). The assumption of justice promotes a critical appropriation of "the Spirit of the Bible" in all readings (Cassuto 1961, p. 13). In other words, the Canon does set limitations on the possibilities of what biblical texts mean across historical, cultural, and geographical contexts. Guided by the assumption of justice, any reading that promotes partial justice cannot be considered a just appropriation of a biblical text.

A closer reading of the Old Testament will privilege the creation, the Abrahamic choice, and prophetic word in a quest to establish the "spirit of the Bible", which will become the guiding the principle on how we interpret other elements within the Bible. What is apparent is the dominance of "justice and righteousness" in these pivotal narratives in the Old Testament. The argument is that the makers of the Old Testament believed that God's plan was to see the world, the entire human race, and the totality of the created universe revolving around justice and righteousness. All other narratives must, therefore, be read and evaluated on their potential contributions to the realization of justice and righteousness throughout the world. It is, therefore, possible to consider the Exodus not just as an exit but rather a rescue from an unjust situation to a situation where justice will become the life principle. God's intervention is not simply for a specific group of people but against an unjust, oppressive, and discriminatory system. The Hebrew slaves are merely an example of exploited and oppressed persons, God is against injustice and intervenes for justice, freedom, and equality (Pacheco 2017, p. 58). Injustice, especially perpetrated by the powerful and sometimes by the majority, has a tendency of rationalizing itself, maybe even made to appear like it is a privilege being extended to the weak and oppressed, and that justice is theological and not simply legal, hence:

*"Conforming to the law is therefore not the best test for justice; the laws themselves may need to be interrogated. The fear of "legitimate manipulation" of justice may be behind the call such as in Exodus 23:2: "You shall not follow a majority in wrongdoing; when you bear witness in a lawsuit, you shall not side with the majority so as to pervert justice." Another text that indicates that justice has a theological foundation is 1 Kings 3:28, which credits Solomon of being just because he had the wisdom of God. This study therefore reckons justice as conforming to laws in a legal and theological way".* (Gunda 2015, p. 22)

The weakness of the assumption of historicity lies in the fact that, whether applied by untrained or trained readers, it strengthens the basis of contesting the historicity and appropriateness of many elements narrated in the Hebrew Bible (Barr 2000, p. 18). Conversely, the strength of the assumption of justice is that it seeks appropriateness of biblical narratives not in their historicity but in their religious-theological ideological contributions to the establishment of harmony among the peoples of the world and by extension to all

creatures of the universe. The Exodus was clearly a central motif in the lives of the people of the Old Testament because it was the basis upon which justice was to be promoted; in it, God had acted decisively against oppression, injustice, and exploitation. The Exodus was not to be the basis for injustice because Israel was not the only beneficiary of God's actions of transferring nations to new territories (Amos 9:7) (Hoffman 1989).

Through the assumption of justice, we could argue that readings that supported apartheid erred in not appreciating the centrality of justice for all and took a pilot project as an exclusive divine project that sought the welfare of one against all others. Similarly, the readings that led to the dismantling of apartheid reached a conclusion that we agree with, yet we also disagree because they gave rise to a class of indigenous oppressors who considered themselves the chosen ones, hence we argue for an assumption of justice that places a premium on justice for all at all times.

## 7. Conclusions

The world continues to struggle with injustice, oppression, and exploitation of one by the other or of the environment by human beings. Unfortunately, the injustice, oppression, and exploitation has been sanitized by an approach to the Bible that is guided by the assumption of historicity that thrives on exclusive privilege in which readers seek to sanitize their positions at the expense of the welfare of the other. The case of apartheid, and its racist, economically exploitative, unjust, and oppressive tendencies shows how the Exodus read through this assumption of historicity became a source of injustice in South Africa and many other places across the universe. Injustice was only considered as such when it was directed against the reading community and similar acts could then be considered as divine when the reading community was now the perpetrator of injustice. "The pains of apartheid and segregation of the colonial regimes were so bitter that the only desirable society was one that would guarantee the freedom and equality of all human beings. The powerful syndicates that control societies are all guilty of equating their happiness and comfort with that of everyone else, including those that they ruthlessly crush into poverty. Indeed, poverty is the by-product of the creation of wealth by these ethnic, social, economic and political syndicates" (Gunda 2015, p. 81). However, Africa is awash with examples of liberators-turned-oppressors and xenophobia sponsors.

This realization made us propose an alternative assumption, the assumption of justice as a profitable way of approaching the religious-theological ideologies of the Old Testament. This assumption considers justice and righteousness to be at the core of the "Spirit of the Bible" and that these are elements that God wills all peoples, powerful or weak, white or brown. This assumption finds expression in the creation theological-ideology (Genesis 1), in the setting apart of Abraham (Genesis 18:19), in the prophets (Amos 9:7), and also in the New Testament, where;

> "*The Jesus' movement was standing up for justice, for there cannot be a true liberation that is not accompanied by justice! While setting at liberty would be regarded as confrontational to the oppressive systems, it was also a call for those joining the movement to commit themselves to upholding justice in their own lives. The wrong things being done by the current powers-that-be could not be practiced by those within the movement. The movement was offering an alternative way of life! It was revolutionary! The greatest commitment for those belonging with the movement was not what they believed in terms of abstract and philosophical doctrines but what they committed themselves to doing in their everyday lives*". (Gunda 2018, p. 63)

**Funding:** This research received no external funding.

**Institutional Review Board Statement:** Not Applicable.

**Informed Consent Statement:** Not Applicable.

**Acknowledgments:** I am grateful for the support of the Phillip Schwarz Initiative of the Alexander von Humboldt Foundation who funded my fellowship stay at the University of Bamberg, Germany.

**Conflicts of Interest:** The author declares no conflict of interest.

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
