# Peer review of "Understanding the Role of the Exodus in the Institutionalization and Dismantling of Apartheid: Considering the Paradox of Justice and Injustice in the Exodus"

_religions, doi:10.3390/rel12080605_

Round 1
Reviewer 1 Report
This article argues that the exodus played a key role both in the establishment and justification of apartheid in South Africa, as well as in the dismantling of this ideology. The essay offers a helpful discussion of the background of apartheid and its development, including its place in theological and social discourse prior to its political enactment. While the essay makes a number of interesting observations, there are some areas that need attention and which would strengthen the article.
The discussion concerning the origins of apartheid and its theological underpinnings is interesting. The discussion of how exodus (and conquest) tropes were used, and the sense of victimisation from Afrikaners, is very insightful. While it seems clear that the exodus (and perhaps also worth differentiating, the conquest) tropes were employed, it may be helpful to situate this use as 'a' foundational experience and text used in the rise of apartheid, rather than 'the' experience (p. 5).
Moving to section 4 and the use of exodus ideas in the fight against apartheid, more is needed in terms of clear examples of how the exodus tradition was actually used in South Africa -- there is little direct evidence presented here when compared with the previous section. Examples are brought in from Zimbabwe and the USA, though it is not clear how these relate to South Africa and apartheid in particular. This section would need significant revision, focusing on the South African context with specific examples of how the exodus was used in dismantling apartheid.
Section 5 focuses on historicity, suggesting that it was a particular historicised reading that allowed for the text to be appropriated by both colonisers and colonised. It is not clear to this reader how historical matters are important here, nor how this particular issue would be important to these interpretive communities. The text may have been appropriated because readers sensed similar contexts and situations, and they would have likely assumed some historicity for the event, but it is unclear how the issue of historicity would be the driving factor (the discussion of critical scholarship is a bit outdated as well, as most contemporary scholars are not concerned solely with historical matters). What might be more helpful would be a discussion on the nature of interpretation, and the situatedness of interpretive endeavours -- what does it mean that both coloniser and colonised could use the same texts for such different ends? Some reflections on hermeneutics and the ethics of interpretation might be helpful in terms of offering a transition to the final section on reading for justice.
While the essay shows adequate research, there is little engagement with current scholarship on Exodus and its reception (e.g., Langston 2005; Baden 2019), nor on postcolonial interpretation, which might have been helpful for placing this particular example in a larger context.
While the essay is written in an engaging and readable manner, there are some issues with style and presentation - grammar and syntax in particular need some attention.
Author Response
- "a" foundational experience as opposed to "the" foundational experience. I understand the concern and will make the necessary adjustment.
- On section 4, I understand that maybe I could find more examples of the use of the exodus by South Africans in their fight against apartheid. However, I disagree on the appropriateness of examples from other places because already in the earlier sections, I have consistently demonstrated that while apartheid was explicitly named so in South Africa, the theology and ideology of separateness could be identified in all other colonies. I, therefore, do not agree that USA and Zimbabwean examples are out of place even though I acknowledge it would be better to include South African examples.
- Section "on historicity" - I understand the reviewer`s suggestions but I also think the reviewer has not fully understood nor appreciated my use of the phrase "the assumption of historicity." In this article, I suggest that one of the things I have observed in the readings of people who may not share the same methodological concerns - literalists, historical critics, postmodern critics including some postcolonialists is the shared "assumption of historicity", which assumes either that narrated stories happened as narrated or to some degree and attach significance to this element as the basis of acknowledging the authority of the Bible or that because some communities have gone through the experiences narrated in some texts, it must mean the text narrates something that must have happened in some form. My point is that across the divide, this "assumption of historicity" plays a role in motivating people to acknowledge the authority of this text hence appropriating it for themselves or to their advantage. While I acknowledge that I might not have accessed some of the suggested scholars and their most recent works on the Exodus, I am not sure if these works will have the effect of altering the argument around the "assumption of historicity". Reading the other reviewers` reports suggests this might not be the case.
- Not being an English native, I understand there might be areas of the manuscript that would need some touch ups and I will try, using also the other reviews, to make these corrections.
Reviewer 2 Report
An important article combining the past and the present, the far-away and the close-by in geography and history. No issue is more relevant than racism in this country —and in the world. No sacred writ has been and is distorted than the Bible; the theme of Exodus is a prime example.
Author Response
Thank you
Reviewer 3 Report
This is a well done and interesting proposal. I recommend removing references to American slavery (lines 190-201, 324-335) as they are tangential to the point of the article.
Additionally, I found typographical errors in the following places:
-line 81 - 1834, however should be 1834. However...
-line 103 - ordained hence should be ordained, hence
-line 112 - called), Revelation should be called) and Revelation
-line 147 - begin a new paragraph at "It is important to observe..."
-line 157 - 1807 was should be 1807 and was
-line 169 - no comma after Great Trek
-line 202 - historicity! A should be historicity, a
-line 208 - figura should be figure
-line 413 - historical critical should be hyphenated
-line 414 - Egypt, yet should be Egypt. Yet
-line 494 - guiding the principle should be guiding principle
-line 524 - promoted, in should be promoted. In
-line 541 - economic exploitative should be economically exploitative
Author Response
Thank you and I will work on the areas you highlighted.
Round 2
Reviewer 1 Report
Appropriate revisions have been made, particularly in the section on the relationship of the Exodus traditions and Apartheid.